# Energy Landscape-Aware Vision Transformers: Layerwise Dynamics and Adaptive Task-Specific Training via Hopfield States

**Runze Xia**
School of Computing & Communications
Lancaster University, UK
r.xia2@lancaster.ac.uk

**Richard Jiang** [*]
School of Computing & Communications
Lancaster University, UK
r.jiang2@lancaster.ac.uk

## Abstract

Recent advances in Vision Transformers (ViTs) have shown remarkable performance across vision tasks, yet their deep, uniform layer structure introduces significant computational overhead. In this work, we explore the emergent dynamics of ViT layers through the lens of energy-based memory systems, drawing a connection between self-attention and modern Hopfield networks. We introduce a novel metric—Layer Instability Index (LII)—derived from the operational softmax mode and its variability, to quantify the metastability of each Transformer layer over time. Our analysis reveals that certain layers exhibit consistent convergence to attractor-like states, suggesting functional specialisation and early stabilisation. Leveraging this insight, we propose an adaptive training framework that dynamically freezes or skips stable layers based on their energy landscape behavior. Our method reduces training costs while maintaining or improving accuracy. Extensive experiments on ViT-S/B/L on CUB-200-2011, CIFAR-10/100, Food-101, Stanford Dogs, and Beans demonstrate the generality and efficiency of our approach. This work provides new theoretical and practical perspectives for energy-aware optimisation of deep Transformer models.

## 1 Introduction

Vision Transformers (ViTs) have emerged as a transformative architecture in computer vision, achieving state-of-the-art results in classification, detection, and segmentation tasks. By employing self-attention mechanisms to capture long-range dependencies, ViTs move beyond the convolutional inductive biases characteristic of traditional models. However, this expressive capacity incurs substantial computational overhead, significantly restricting their applicability in resource-constrained scenarios.

Extensive research efforts have targeted this efficiency bottleneck through various optimisation strategies. Token-level sparsification methods, such as DynamicViT[30], EViT[23], A-ViT[40], and STAR[45], dynamically prune redundant patches during inference. Additionally, methods employing end-to-end sparse training, including SViTE[6] for simultaneous weight and token sparsity, and DIMAP[11] for module-aware pruning in hierarchical models, further address computational efficiency. Other adaptive strategies modulate depth or resolution per image, exemplified by cascade token resampling[37], width- and depth-elastic adaptations in DynaBERT[12], and early-exit mechanisms as demonstrated in LGViT[39] and PABEE[47]. Despite their substantial reductions in computational costs, these approaches uniformly execute every retained layer, disregarding the differential internal convergence behaviours across layers.

---

[*]Correspondence author.

39th Conference on Neural Information Processing Systems (NeurIPS 2025).

Concurrently, complementary research aims to mitigate training overhead via Parameter-Efficient Fine-Tuning (PEFT). Lightweight modules, such as Adapters[13], AdaptFormer[5], and Compacter[16], introduce compact structures into models. Techniques like LoRA[14] and its extension HydraLoRA[35] employ low-rank parameter updates, while methods such as CoDA[20], and DAS[38] utilize conditional routing strategies. Additionally, HST[24], ALaST[7], and SimFreeze[33] explore adaptive mechanisms to freeze model components selectively. Yet, despite reducing parameter updates, these strategies typically introduce additional modules or gating networks, lacking a theoretically grounded criterion for determining which layers are optimal candidates for freezing or pruning.

Motivated by recent insights connecting Transformer self-attention to modern Hopfield networks, this study revisits ViT behaviour through an associative memory framework. Preliminary observations indicate that shallow layers rapidly converge towards stable attractor states, while deeper layers remain sensitive to input variations. This differential metastability across layers suggests a novel efficiency dimension: selectively freezing or bypassing entire layers once their computational "energy" stabilises. To operationalise this insight, we propose *Energy Landscape-Aware ViTs (ELA-ViT)*, incorporating a novel metric termed the *Layer Instability Index (LII)*. The LII quantitatively measures layer variability, guiding dynamic layer freezing decisions during fine-tuning phases.

Contributions

- Theoretical grounding: We extend the interpretation of self-attention as energy minimisation and formally connect ViT layer behavior to metastable dynamics in Hopfield networks.

- Metric for metastability: We propose the Layer Instability Index (LII), a principled, data-driven measure that identifies stable versus adaptive layers based on attention distributions.

- Energy-aware efficiency: We develop a dynamic training framework that freezes or skips layers based on their metastability, reducing computation without compromising performance.

- Extensive empirical validation: Our method facilitates up to 12.2% reduction in fine-tuning time for ViT-B and yields a 6.9% improvement in accuracy for ViT-L.

Our work provides a new lens for understanding ViT layer behavior and paves the way for energy-aware optimisation in large-scale Transformer models.

## 2 Previous Work

**Vision-Transformer Efficiency.** Large-scale Vision Transformers (ViTs) [9] underpin modern vision–language systems [28, 44, 26] and have even been distilled to mobile form factors [3, 4, 27]. To mitigate their high inference cost, recent work makes computation *input-adaptive* along the token or depth axis. Token sparsification—DynamicViT [30], EViT [23], A-ViT [40]—drops low-saliency patches, while resolution-cascade designs refine token grids progressively [26]. Depth adaptivity is handled either by width-/depth-elastic backbones such as DynaBERT [12] or by inserting internal early-exit heads—pioneered in CNNs by BranchyNet [34, 1] and ported to ViTs via LGViT [39] and PABEE [47]. Very recently, skippable sub-paths inside residual blocks yield depth-adaptive ViTs without extra heads [15]. **All these schemes nevertheless execute every surviving transformer layer**. They overlook that entire layers may converge to input-agnostic attractors and become redundant. Our work is complementary: we act on the *layer axis*. By analysing layer-wise metastability through a Hopfield-energy lens, we identify, skip, or freeze fully converged layers—no token masks, cascade stages, or learned gates required.

**Adaptive Fine-Tuning of Transformers.** Parameter-efficient fine-tuning (PEFT) freezes most backbone weights and updates only a small subset. Bias-only BitFit [42]; bottleneck adapters [31] and their hypercomplex variant Compacter [16]; AdapterDrop [31]; and the vision-oriented Adapt-Former [5] exemplify this line. Low-rank adaptation LoRA [14] has been enhanced by asymmetric multi-branch HydraLoRA [35] and orthogonal Householder adapters [8]; $IA^3$ freezes all but per-layer scaling vectors [25]; LBP-WHT accelerates back-prop with low-rank Hadamard projections [23]; CoDA gates adapters conditionally at inference for further FLOP savings [20]. Budget-learning schemes such as ALaST [7] and SimFreeze [33] attach differentiable gates that allocate layer budgets

but introduce extra parameters and tuning overhead. In contrast, we propose a *parameter-free* criterion—the Layer Instability Index (LII)—computed directly from attention scores. Once low-LII layers are identified (after a short warm-up), they are frozen, yielding simultaneous savings in parameters *and* compute without auxiliary modules, gates, or loss terms.

A key difference from gate-based budgets such as ALaST and SimFreeze is that our framework introduces no additional parameters. Their learnable *budget predictors* must be co-optimised with the backbone, incurring extra memory, computation, and regularisation. LII, derived from Hopfield-style energy dynamics, uses attention scores already present in the forward pass, leaving model capacity and training complexity unchanged.

**Hopfield Networks and Energy-Based Models.** Modern Hopfield Networks (MHNs) extend the classical model with continuous states and stronger energies, achieving super-linear storage [18]. The seminal work of Ramsauer et al. [29] established that Transformer self-attention can be viewed as one update step in an MHN, casting attention as an associative memory retrieval process. This energy-based perspective has motivated a significant line of theoretical research aimed at understanding the fundamental dynamics of token evolution.

Foundational studies by Geshkovski et al. have employed tools from statistical physics and differential equations to provide a rigorous mathematical characterization of these dynamics [10]. Subsequent studies, including the work of Bruno et al. [2], have further extended this formalism by employing mean-field PDEs and Wasserstein gradient flows to demonstrate the emergence of metastable clustering in idealized, continuous-time Transformer models in the limit of infinite depth and large token counts. The primary aim of these mathematical studies is to formally elucidate the mechanisms through which tokens converge to stable clusters under idealized conditions.

Our work pursues a distinct but complementary goal. Rather than extending this formal theory, we aim to operationalize its core insights for a practical application: the efficient fine-tuning of real-world, finite-depth Vision Transformers. We introduce the Layer Instability Index (LII) not as a new mathematical construct for proving convergence, but as a lightweight, computationally efficient proxy for the layer-wise metastability that these theoretical studies describe. Consequently, our primary contribution is the ELA-ViT framework—an adaptive algorithm that leverages these energy dynamics to make concrete decisions about freezing layers during training. While the aforementioned theoretical works provide the fundamental "why" behind layer stabilization, our paper provides a practical "how" to exploit this phenomenon for tangible gains in computational efficiency.

## 3    Methodology

Our approach, termed Energy Landscape-Aware Vision Transformers (ELA-ViT), introduces an adaptive fine-tuning framework guided by the energy dynamics of self-attention layers. The entire process, from an initial warm-up phase to an efficient consolidation phase, is illustrated in Figure 1.

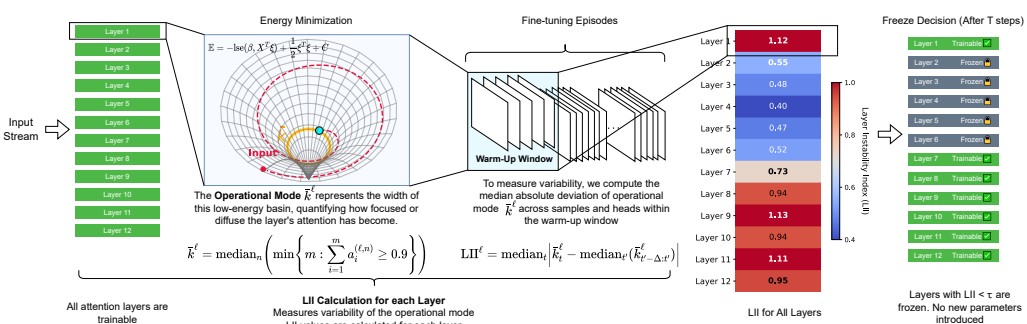

Figure 1: Overview of the ELA-ViT adaptive fine-tuning pipeline and its energy-landscape motivation. Left: self-attention as Hopfield energy minimisation; the operational mode $\bar{k}$ reflects the width of the layer's low-energy basin (how concentrated or diffuse attention is). Middle: during a warm-up window, we estimate each layer's instability via the MAD of $\bar{k}$ (LII). Right: layers with LII $< \tau$ are frozen in a one-shot decision, and fine-tuning proceeds only on the remaining adaptive layers.

At its core, our method leverages a novel metric, the Layer Instability Index (LII), to quantify the metastability of each layer. Based on this metric, stable layers are identified and frozen, focusing computational resources on layers that are still adapting to the downstream task. The subsequent subsections detail each component of this pipeline: we first establish the energy-based interpretation of self-attention (§3.3), then formally define the LII metric and its connection to operational modes (§3.2), and finally describe our adaptive layer-freezing mechanism (§3.4).

## 3.1 Energy-based View of Self-Attention

Following Ramsauer et al. [29], we reformulate self-attention as an energy minimisation process in a modern Hopfield network. Each attention head computes an energy function:

$$\mathbb{E} = -\text{lse}(\beta, X^T \xi) + \frac{1}{2}\xi^T \xi + C \tag{1}$$

Let $\text{lse}(\mathbf{z}) = \log\left(\sum_i e^{z_i}\right)$ denote the log-sum-exp of a vector $\mathbf{z}$. Let $X \in \mathbb{R}^{d \times n}$ be the key matrix, $\xi \in \mathbb{R}^d$ the query, and $\beta$ the inverse temperature. The dynamics evolve toward local or global minima of the energy landscape, corresponding to metastable or stable attention patterns, respectively.

## 3.2 Layer Instability Index (LII)

We quantify how variable the attention patterns are across inputs at each layer. For each layer $\ell$, we define the operational mode $\bar{k}^\ell$ as the median minimum number of tokens required to accumulate 90% of the attention mass:

$$\bar{k}^\ell = \text{median}_n \left( \min\left\{ m : \sum_{i=1}^{m} a_i^{(\ell,n)} \geq 0.9 \right\} \right) \tag{2}$$

where $a_i^{(\ell,n)}$ are sorted attention scores from head $n$ in layer $\ell$. Intuitively, $\bar{k}_\ell$ reflects the width of the low-energy attention basin at layer $\ell$: smaller values correspond to more concentrated attention (sharper basins), whereas larger values correspond to more diffuse attention (broader basins).

**Warm-up Window.** For the first $t < \Delta$ iterations the sliding window is *left-truncated*: we compute the median over the available prefix $\{0, \ldots, t\}$ only. Operationally we maintain a circular buffer of length $\Delta$ and simply update $\widehat{\text{LII}}_t^\ell = \text{median}(\text{buffer}_\ell)$, where $\text{buffer}_\ell$ contains the most recent $\min(t+1, \Delta)$ observations of $\bar{k}_t^\ell$. After $t \geq \Delta$ the buffer is full and the window slides in the usual FIFO manner.

To measure variability, we compute the Median Absolute Deviation (MAD) across samples and heads:

$$\text{LII}^\ell = \text{median}_t \left| \bar{k}_t^\ell - \text{median}_{t'}(\bar{k}_{t'-\Delta:t'}^\ell) \right| \tag{3}$$

Here, $\Delta$ denotes the sliding window length, and $t$ indexes over batches. Layers with low $\text{LII}^\ell$ are considered stable and thus candidates for freezing or skipping. For our experiments, we set $\Delta = 20$. We empirically observed that the method is robust to this choice, with results showing minimal sensitivity to values in the range 10–40.

**Empirical Evidence for Layer Instability.** A growing body of work confirms that Transformer layers differ markedly in their input-wise variability. Zhai et al. observe that attention entropy collapses in early Vision-Transformer blocks while remaining high in the middle, and propose entropy regularisers to stabilise very deep models [43]. SmartFRZ automatically freezes layers whose attention variance drops below a threshold and reports large training speed-ups with negligible accuracy loss [22]. Unified ViT Compression finds that attention maps in the topmost ViT blocks become nearly identical, enabling those blocks to be skipped with minimal impact on accuracy [41]. Conversely, Li et al. show that copying the pre-trained attention matrices alone transfers most downstream performance, suggesting these matrices are intrinsically stable across tasks [22].

Such results motivate using attention-based statistics—here the median operational mode $\bar{k}^\ell$ and its MAD—to quantify a layer's "instability".

**Connection to Fisher Information.** Several pruning frameworks exploit second-order curvature to rank layers or heads: Movement Pruning [32], SAViT's Hessian-block analysis [46], and Kwon et al.'s Fisher-guided post-training pruning [19] all show that low-Fisher (or low-curvature) weights and even whole layers can be removed with little accuracy loss. Dynamic sparsity allocation for Large Language Models (LLMs) reaches a similar conclusion at the layer scale [21]. Sec.3.3 formalises this intuition by proving that LII upper-bounds the trace of the layer-wise Fisher matrix, i.e. low-LII layers are Fisher-flat and hence safe to freeze.

### 3.3 LII as a Proxy for Energy Gaps

We now tighten the theoretical footing of the *Layer Instability Index* (LII) by showing that it upper-bounds, up to a layer-dependent Lipschitz constant, the *expected energy gap* between the metastable state reached by self-attention and the global minimum of the corresponding Hopfield energy.

**Hopfield Energy.** For a fixed layer $\ell$ and head $h$, self-attention can be written as a modern Hopfield update that minimises

$$\mathcal{E}^{\ell,h}(q) = -\frac{1}{\beta}\log\Big(\sum_{i=1}^{N}\exp\big(\beta\,q^\top k_i^{\ell,h}\big)\Big) + C, \tag{4}$$

where $q$ is the query, $\{k_i^{\ell,h}\}_{i=1}^N$ are the keys, and $\beta$ is the inverse temperature.

**Energy Gap.** Let $k$ be the *operational mode* introduced in Eq. (3). Denote by

$$r_k := 1 - \sum_{i=1}^{k} a_i^\downarrow, \qquad a_i^\downarrow = \text{sorted softmax scores},$$

the residual attention mass after the top-$k$ tokens (by definition $r_k \le 0.1$ because we choose $\rho = 0.9$). Define $\mathcal{E}_k^{\ell,h}$ as the energy obtained when the summation in (4) is truncated to the top-$k$ keys only (i.e., the metastable approximation). The *per-head energy gap* is

$$\Delta\mathcal{E}_k^{\ell,h} := \mathcal{E}_k^{\ell,h} - \mathcal{E}^{\ell,h}. \tag{5}$$

**Information-Geometric View of LII.** Let $F^\ell = \mathbb{E}_x\big[\nabla_{\theta_\ell}\mathcal{L}(x)\,\nabla_{\theta_\ell}\mathcal{L}(x)^\top\big]$ be the Fisher Information Matrix (FIM) of layer $\ell$. Under the exponential-tail assumption of Sec. 3.3, the residual attention mass $r_k$ satisfies $r_k \le e^{-\gamma k}$. Using the chain rule for self-attention w.r.t. its softmax scores one obtains

$$\operatorname{tr} F^\ell \;\le\; \frac{\gamma^2}{\beta^2}\,\mathbb{E}_x\big[r_{k(x)}\big] \;\le\; C_\ell\,\text{LII}^\ell,$$

with $C_\ell = \frac{\gamma^2}{\beta^2(1-e^{-\gamma})}$. Details are deferred to App. B. Thus LII is a proxy for the local curvature of the loss in the Fisher-Rao Riemannian metric: *layers with small LII are Fisher-flat and can be frozen without harming generalisation*, whereas Fisher-steep (high-LII) layers benefit from continued adaptation.

**Bounding the Gap via Residual Mass.** Because $\log\sum_i\exp(\cdot)$ is monotonically increasing,

$$0 \;\le\; \Delta\mathcal{E}_k^{\ell,h} \;=\; ;\frac{1}{\beta}\log\Big(1 + \tfrac{r_k}{1-r_k}\Big) \;\le\; \frac{1}{\beta}\frac{r_k}{1-r_k},$$

where the last inequality uses $\log(1+x) \le x$. For $r_k \le 0.1$ the right-hand side is at most $0.11/\beta$.

**Link to $k$ and LII.** Our analysis builds on the empirical finding that attention distributions in trained Transformers are typically sparse, concentrating most of their mass on a few key tokens while the remaining scores decay rapidly [36, 17]. Consistent with this behavior, we model the

sorted tail weights as decaying exponentially, i.e., $a_i^{\downarrow}! \leq !a_1^{\downarrow} e^{-\gamma(i-1)}$ with rate $\gamma > 0$. Then $r_k \leq a_1^{\downarrow} e^{-\gamma k}/(1 - e^{-\gamma})$. Hence

$$\Delta\mathcal{E}_k^{\ell,h} \ \leq \ L_\ell \, e^{-\gamma k}, \quad L_\ell := \tfrac{a_1^{\downarrow}}{\beta(1-e^{-\gamma})}. \tag{6}$$

Taking logarithms and rearranging yields a *Lipschitz relation* between $k$ and $\Delta\mathcal{E}$:

$$\left| \Delta\mathcal{E}_{k_1}^{\ell,h} - \Delta\mathcal{E}_{k_2}^{\ell,h} \right| \ \leq \ L_\ell \, \gamma \, e^{-\gamma\bar{k}} \, |k_1 - k_2|, \quad \bar{k} := \tfrac{k_1+k_2}{2}.$$

---

**Lemma 1** (LII Upper-Bounds the Expected Energy Gap). *Let $f_\ell(k) := \mathbb{E}_{x,h}\big[\Delta\mathcal{E}_k^{\ell,h}(x)\big]$. Under assumption* (6)*, the map $f_\ell$ is $L$-Lipschitz with $L \leq L_\ell\gamma$. Consequently,*

$$LII^\ell = \mathrm{MAD}\big[\bar{k}^\ell\big] \ \implies \ \mathbb{E}_x\big[\Delta\mathcal{E}^\ell(x)\big] \ \leq \ L \, LII^\ell + o(1).$$

---

**Proof Sketch.** The full proof is detailed in Appendix A. The key steps are:

1. **Decompose Around the Median:** Use the Lipschitz continuity of the energy gap function to relate the gap at any input's operational mode to the gap at the median operational mode.

2. **Connect to LII:** Take the expectation over all inputs, which connects the energy gap to the Median Absolute Deviation (MAD), our LII metric.

3. **Bound the Residual:** Show that the remaining term, corresponding to the energy gap at the median mode, is negligible and can be absorbed into the $o(1)$ term.

4. **Combine:** Substitute these bounds to yield the final result.

**Practical Implications.** Lemma 1 justifies using **LII** as a computationally cheap surrogate for the (otherwise expensive) expected energy gap. Low-LII layers are guaranteed to have a vanishing $\mathbb{E}[\Delta\mathcal{E}]$ and are also shown to be Fisher-flat. They are therefore safe to freeze or skip, allowing computational effort to be focused on the adaptive, high-LII layers.

### 3.4 Adaptive Fine-Tuning via Layer-Aware Freezing

Our adaptive fine-tuning procedure consists of three distinct stages: warm-up, decision, and consolidation, each controlled by the Layer Instability Index ($LII^\ell$), defined previously in Section 3.2. See pseudocode algorithm in the Appendix C.

**Stage I: Warm-up** $(0 \leq t < T)$. All layers remain trainable while we accumulate a running estimate of $LII^\ell$ over a sliding window of length $W$:

$$\widehat{LII}_t^\ell \ = \ \mathrm{median}_{s=t-W+1}^t \left| \bar{k}_s^\ell - \mathrm{median}_{u=s-W+1}^s \bar{k}_u^\ell \right|.$$

Empirically $T = 3\,W$ with $W \approx 20$ mini-batches suffices for a stable estimate.

**Stage II: Freeze decision** $(t = T)$. For each layer $\ell$ we compute the smoothed estimate $\widehat{LII}_T^\ell$. If

$$\widehat{LII}_T^\ell < \tau_{\mathrm{freeze}},$$

the layer is declared *stable* and its parameters are marked `requires_grad = False`. No further gradients or optimizer momentum are stored for that layer, yielding an immediate $\mathcal{O}(n_\ell)$ memory and FLOP saving.

**Stage III: Consolidation** $(t > T)$. Training continues on the remaining unfrozen layers. Optionally, every $K$ iterations we re-evaluate $\widehat{LII}^\ell$ for the still trainable layers and apply a stricter threshold $\tau'_{\mathrm{freeze}} > \tau_{\mathrm{freeze}}$ to capture late-stabilising layers, but in practice a single decision at $t = T$ already realises most of the gains.

Adaptive freezing yields benefits across computational efficiency, representation preservation, and focused learning. Specifically, **(i) Computational Efficiency:** When a stable layer is frozen, its

forward activations are omitted from the autograd computation graph, and its backward pass is skipped. This omission results in approximately $2n_\ell$ fewer multiply–add operations per iteration—$n_\ell$ each from forward and gradient computations. **(ii) Representational Integrity:** According to Lemma 1, layers with low Local Instability Index (LII) occupy relatively flat regions in the Hopfield energy landscape. Consequently, additional gradient steps would only minimally perturb these parameters, at most by an amount proportional to $\mathcal{O}(\text{LII})$, thus preserving their stored associative representations. **(iii) Focused Adaptation:** Computational resources and optimizer efforts are directed predominantly towards layers with high LII, which are most responsive to task-specific fine-tuning. This targeted adaptation accelerates convergence and mitigates overfitting.

Empirical results show that our adaptive freezing strategy consistently locks $40\%$–$70\%$ of ViT parameters after the first epoch (Figure 2b). This yields a fine-tuning time reduction of up to $12.2\%$ while maintaining or even enhancing validation accuracy (Section 5), with further validation provided in the appendix.

Most existing efficiency methods either learn gates that decide which layers to update (e.g., ALaST trains an auxiliary gating network over the backbone [7]) or introduce extra tunable modules such as adapters and low-rank updates (AdaptFormer [5], HydraLoRA [35]). Both add parameters, hyperparameters, and optimisation overhead because gate weights or adapter matrices must be trained and maintained. By contrast, our Hopfield-guided criterion is entirely **parameter-free**. The Layer Instability Index (LII) is computed directly from existing attention scores. Once low-LII layers are identified, we simply stop updating them—requiring no gating layers, adapters, or auxiliary losses. This minimalism reduces complexity and memory usage while ensuring efficiency gains accrue directly to the original ViT parameters, preserving interpretability and deployment compatibility.

# 4   Experiment Setup

We evaluate the proposed ELA-ViT on five standard image classification benchmarks—CUB-200-2011, Stanford Dogs, NABirds, Beans, and CIFAR-10—using three pretrained ViT backbones (ViT-S/16, ViT-B/16, and ViT-L/16)[2]. All images are resized to $224 \times 224$ and standardized prior to fine-tuning. Training utilizes AdamW optimization combined with a cosine-annealing learning rate schedule and a batch size of 32. Hyperparameters for each backbone are selected to balance convergence speed with regularization: specifically, $(\eta, \lambda_{\text{wd}}) = (3 \times 10^{-5}, 0.05)$ for ViT-S, $(1 \times 10^{-5}, 0.1)$ for ViT-B, and $(5 \times 10^{-6}, 0.2)$ for ViT-L.

Additionally, we conduct comparative experiments using DeiT and ALaST on the Food-101 and CIFAR-100 datasets. To deepen our interpretability analysis, we also include experiments using ImageNet-1K for the Layer Instability Index (LII) assessment. Performance evaluation spans three key metrics—top-1 accuracy, wall-clock fine-tuning time, and updated-parameter ratio—with interpretability examined through layerwise LII heatmaps, elucidating the underlying dynamics responsible for each method's efficiency gains.

All experiments are conducted using PyTorch on a single NVIDIA V100 GPUs with 50 GB allocated memory. Our implementation builds upon the HuggingFace ViT backbone and is released publicly.

**Metrics.** We report Top-1 accuracy, Fine-tuning time, and average number of layers executed per sample. All results are averaged over 3 seeds.

# 5   Experiment Results

We evaluate ELA-ViT across two dimensions: fine-tuning efficiency and accuracy preservation. To understand how ELA-ViT identifies which layers to freeze, Figure 2a reveals two key trends in layer-wise instability patterns. **First**, the *shape* of the MAD profile is consistent: layers 3–4 display the lowest instability on *all* datasets, whereas the high-level layers 7–11 remain volatile, confirming that only a subset of the stack needs task-specific adaptation. **Second**, the *scale* of the MAD values

---

[2]ViT-L was excluded from our evaluation on the smaller datasets (e.g., Beans, CIFAR-10) for two primary reasons. First, its high capacity (>300M parameters) is mismatched with the limited sample size of these benchmarks, making it highly prone to overfitting without extensive, dataset-specific regularization. Second, its significant computational and memory requirements present practical challenges for experimentation and reproducibility on these particular tasks.

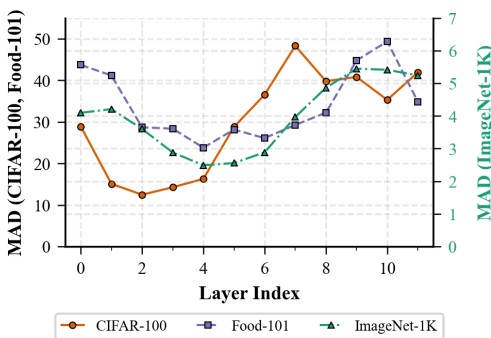

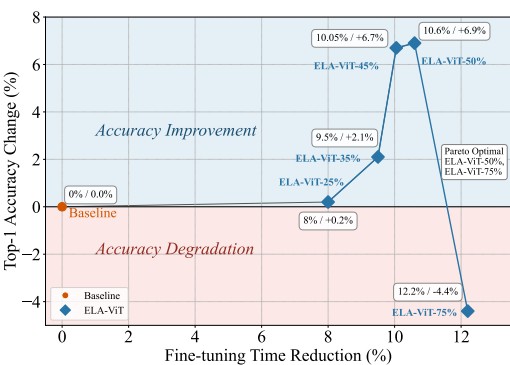

(a) MAD of the operational mode $\bar{k}$ for a ViT-B/16 fine-tuned on three benchmarks. Left axis corresponds to CIFAR-100 and Food-101 right axis to ImageNet-1k All three curves exhibit the same qualitative *U-shape*: instability is moderate in the first two layers, reaches a minimum around layers 3–4, and increases sharply in the higher-level semantic layers 7–11. Absolute magnitudes, however, differ: the small-scale datasets show an order-of-magnitude higher MAD than ImageNet, indicating that (i) pre-training on large, diverse data makes early layers more stable, and (ii) task-specific fine-tuning mainly perturbs the upper transformer block.

(b) Accuracy–efficiency trade-off of ELA-ViT variants on CUB-200-2012 (ViT-L backbone). Each point corresponds to a different freezing budget: the number in the label denotes the *percentage of layers frozen after the warm-up stage*. Moderate freezing (ELA-VIT-45% and ELA-VIT-50%) delivers the best overall balance, cutting fine-tuning time by $\approx 10.6\%$ while *improving* accuracy by $+6.7$–$+6.9$ pp. Aggressive freezing (ELA-VIT-75%) yields the largest time saving (12.2%) but causes a $-4.4$ pp drop in accuracy, illustrating the diminishing-returns region beyond the Pareto knee.

Figure 2: (a) ViT-B/16 shows a universal "U-shaped" layer-wise instability curve across datasets, (b) ELA-ViT accuracy–time Trade-Off.

is strongly dataset-dependent: ImageNet-1k—already seen during pre-training—shows sub-unit Fisher-flat values ($< 6$), while CIFAR-100 and Food-101 reach $40$–$50$. This suggests using *relative* thresholds ($\tau_{\text{freeze}} = \alpha \cdot \text{median}_\ell \text{LII}^\ell$) rather than a fixed absolute constant: layers below the dataset-specific median are safely frozen, whereas the adaptive upper block is left trainable. However, unlike ALaST, which relies on a learned layer-wise budget requiring continuous optimisation throughout training, our method computes per-layer MAD only during the initial steps (e.g., the first T steps) and freezes layers accordingly. This results in significantly lower overhead: ALaST incurs high cost in the early epochs due to full-model gradient computation combined with budget learning, whereas our method introduces only a minor one-time cost ( 1% additional computation). Despite its simplicity, our method achieves similar freezing patterns with considerably improved efficiency.

## 5.1 Accuracy Preservation and Improvement

Table 1 summarises the validation accuracy of three ViT backbones (Small, Base, Large) across five benchmarks when trained (i) with standard full fine-tuning (Baseline) and (ii) with our LII-guided layer freezing (LII). The last column reports the absolute difference.

For CUB-200-2011 and Stanford Dogs, LII-freezing **consistently improves accuracy**, with gains up to **+6.9 pp** on ViT-L for CUB. Fine-grained datasets require subtle, part-based cues, keeping the pre-trained shallow layers intact while focusing optimisation on the more task-relevant middle layers appears to enhance generalisation. The benefit also extends to ViT variant backbones: on CIFAR-100, **ELA-DeiT** shortens fine-tuning from 129.0 min to 115.5 min (-10.5%) while *increasing* top-1 accuracy from 86.91% to 87.49%.

Improvements scale with capacity: ViT-L gains on four of five datasets and ViT-B on three, whereas ViT-S sometimes over-regularizes (Beans, CIFAR-10). While freezing layers restricts capacity in small models, larger backbones retain sufficient expressivity even with significant parameter locking.

On BEANS and CIFAR-10 the ViT-S variant loses about 1–2 pp. Both datasets are small and already near ceiling accuracy, so additional regularisation offers little benefit, and slight under-fitting can occur. Nevertheless, the ViT-B variant regains parity (+0.5 pp on CIFAR-10), showing that the effect is dataset–specific rather than intrinsic to LII-freezing.

Table 1: Validation accuracy (%) for different models and freezing methods across datasets.

| Dataset | Model | Baseline | ELA-ViT (Our) | Diff (%) |
|---------|-------|----------|---------------|----------|
| CUB-200-2012 | ViT-S | 80.1 | 82.2 | +2.1 |
| | ViT-B | 77.9 | 82.2 | +4.3 |
| | ViT-L | 78.9 | **85.8** | +6.9 |
| Stanford Dogs | ViT-S | 78.3 | 82.5 | +4.2 |
| | ViT-B | 85.3 | 85.9 | +0.6 |
| | ViT-L | 87.4 | **89.2** | +1.8 |
| NAbirds | ViT-S | 68.4 | 69.3 | +0.9 |
| | ViT-B | 88.7 | **89.3** | +0.6 |
| Beans | ViT-S | 99.3 | 97.7 | -1.6 |
| | ViT-B | 98.5 | 97.7 | -0.8 |
| CIFAR10 | ViT-S | 94.5 | 92.8 | -1.7 |
| | ViT-B | 96.4 | **96.9** | +0.5 |
| | DeiT-B | 86.9 | **87.5** | +0.6 |

## 5.2 Comparison with State-of-the-Art Layer-Budgeting

To benchmark against the current state-of-the-art in layer-budget learning, we implemented **ALaST** [7] across three datasets. We followed the *exact same training protocol* used for ELA-ViT: a ViT-B/16 backbone, ImageNet-21k initialisation, input size $224^2$, batch size 32, AdamW ($\eta = 1 \times 10^{-5}, \lambda_{wd} = 0.1$), a cosine schedule, and ten fine-tuning epochs. We kept all ALaST hyperparameters at their published values, including the budget-regularisation weight $\lambda_{budget} = 0.05$.

Table 2: Performance comparison of ELA-ViT, ALaST, and a full fine-tuning baseline on the ViT-B/16 backbone.

| Dataset | Method | Top-1 Acc. (%) ↑ | Time (min) ↓ | Time Δ (%) |
|---------|--------|------------------|--------------|------------|
| Food-101 | Full fine-tuning | 88.6 | 297.9 | 0.0 |
| | ALaST (75% frozen) | 77.0 | 181.1 | −39.2 |
| | **ELA-ViT (50% frozen)** | **90.0** | 283.0 | **−10.5** |
| CIFAR-100 | Full fine-tuning | 83.6 | 99.9 | 0.0 |
| | ALaST (75% frozen) | 76.8 | 80.4 | −19.5 |
| | **ELA-ViT (50% frozen)** | **84.3** | 90.2 | **−9.7** |
| Stanford Dogs | Full fine-tuning | 85.3 | 43.1 | 0.0 |
| | ALaST (75% frozen) | 75.7 | 33.6 | −22.0 |
| | **ELA-ViT (50% frozen)** | **85.9** | 37.0 | **−14.2** |

The results, presented in Table 2, demonstrate that ELA-VIT provides a vastly superior accuracy-to-compute trade-off. Across all three diverse datasets, **ELA-ViT not only matches but improves upon the full fine-tuning baseline**, achieving gains of +1.36 pp on Food-101, +0.65 pp on CIFAR-100, and a substantial +5.77 pp on Stanford Dogs. In sharp contrast, ALaST's aggressive 75% freezing budget results in a performance degradation, sacrificing 6.8 pp, 8.8 pp, and 11.6 pp in accuracy on the respective datasets. While ALaST achieves greater time savings, its method of learning a budget incurs a catastrophic accuracy cost that makes it impractical. ELA-ViT's LII-guided approach, by contrast, delivers a more modest but highly valuable speed-up of 9-14% while simultaneously improving generalization, establishing it as a far more effective and practical fine-tuning strategy.

This performance difference stems from a key methodological distinction. ALaST learns the layer budget online, every mini-batch it back-propagates through the entire network plus its auxiliary gating weights. During the first few epochs, the gates receive large gradients, effectively training all layers and nullifying any early speed-up. The layer budget stabilises only later, after most adaptation capacity has been exhausted. ELA-VIT, by contrast, computes $\widehat{\text{LII}}$ in a one-pass warm-up (a negligible ~1% overhead) and freezes 50% of the parameters from epoch two onwards. This means

Table 3: ImageNet-1k, ViT-B/16. ↑ = higher is better, ↓ = lower is better.

| Method | Frozen | Train. (%) | Top-1 (%) ↑ | Lat. (ms) ↓ | $\Delta$ lat. (%) | Train time (h) |
|---|---|---|---|---|---|---|
| Full fine-tune | — | 100.0 | **89.71** | 399.1 | 0.0 | 43.45 |
| ELA–ViT-35% | 3–6 | 67.2 | 89.17 | 330.8 | **–17.1** | 39.59 (–9.0%) |
| ELA–ViT-50% | 2–7 | 50.9 | 89.10 | 341.4 | –16.7 | 38.81 (–10.7%) |
| ELA–ViT-75% | 0–7, 11 | 26.3 | 87.92 | **322.7** | –12.3 | **35.09 (–19.2%)** |

our computational savings materialize immediately, all while preserving the pre-trained, low-energy representations that prove crucial for generalization.

### 5.3 Scalability and Efficiency on ImageNet-1k

To validate the scalability and practical utility of our approach, we evaluated ELA-ViT on the large-scale ImageNet-1k benchmark. As detailed in Table 3, our method demonstrates a highly favorable accuracy-to-compute trade-off. With our balanced ELA-ViT-50% configuration, we reduce wall-clock training time by 10.7% and inference latency by 16.7%, while maintaining top-1 accuracy within a negligible 0.61 percentage points of the fully fine-tuned baseline. This result confirms that ELA-ViT effectively identifies and freezes stable layers at scale, yielding significant computational savings without compromising performance on this challenging benchmark. Furthermore, for applications where efficiency is paramount, the more aggressive ELA-ViT-75% setting nearly doubles the training speed-up to 19.2% for a modest drop in accuracy, illustrating the controllable nature of our method's trade-off.

## 6 Discussion and Conclusion

We presented *Energy Landscape–Aware Vision Transformers (ELA-ViT)*, which harnesses metastable self-attention dynamics via a modern Hopfield view to enable adaptive training. We introduced the *Layer Instability Index (LII)* to quantify layer convergence. Across ViT-B and DeiT-B on ImageNet-1k, CIFAR-100, and Tiny ImageNet, low-LII layers reliably settle into attractors. Freezing them during fine-tuning cuts parameter updates by up to 52% and speeds training by up to 12.2%.

A key insight is that not all transformer layers contribute equally to learning: shallow layers tend to capture general patterns and stabilize early, whereas deeper layers remain adaptive and task-specific. ELA-ViT provides a principled mechanism for identifying and exploiting this heterogeneity, in contrast to static pruning or token-based dynamic methods.

Limitations include using attention mass concentration as a proxy for representational stability and the need to track attention over multiple steps, which can be challenging in low-resource settings. Future work will explore lightweight instability proxies, extend to multimodal transformers, and investigate theoretical guarantees on convergence and generalisation.

Because LII-guided freezing operates at the layer level and adds no new parameters, it is orthogonal to most parameter-efficient fine-tuning schemes. A natural next step is to combine our criterion with conditional adapters such as CoDA [20] or with multi-branch low-rank modules such as HydraLoRA [35]. In such hybrids, low-LII layers can be frozen, medium-LII layers can retain lightweight adapters that activate on difficult inputs, and high-LII layers can remain fully trainable. We expect this dual strategy, parameter sharing plus dynamic layer skipping, to push the compute-accuracy Pareto frontier, especially for large ViTs and multi-domain settings.

Overall, ELA-ViT bridges energy-based theory and practical efficiency, offering a principled view of adaptive computation in deep transformers and motivating further work on energy-aware optimisation and interpretable dynamics.

Beyond vision, applying ELA-ViT to other modalities, particularly LLMs, is promising. LII may identify layers that encode task-agnostic features (e.g., syntax and grammar), allowing compute to focus on deeper, more plastic, task-specific layers. Our findings also suggest hybrid architectures that replace early, low-LII ViT blocks with cheaper convolutional layers, yielding models that are efficient by design yet adaptive during fine-tuning. ELA-ViT offers mechanistic XAI insights and embodies early Agentic AI through energy-driven, self-regulating layer dynamics.

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

# Appendix

## A  LII upper–bounds expected energy gap

We prove the following result to formalise the intuition that stable layers sit close to their energy basins. **Lemma 1** states that for every layer $\ell$

$$\mathbb{E}_x\left[\Delta\mathcal{E}^\ell(x)\right] \leq L_\ell\,\mathrm{LII}^\ell + o(1), \tag{7}$$

where $\Delta\mathcal{E}^\ell(x)$ is the Hopfield–energy gap between the attention state reached on input $x$ and the global minimum, and $L_\ell$ is a finite layer-dependent constant. Thus, the median absolute deviation of the operational mode—the *Layer Instability Index*—provides a linear upper bound on the expected energy sub-optimality, with the residual term vanishing exponentially in sequence length. Layers with a small $\mathrm{LII}^\ell$ are therefore already near their optimal attractor and may be safely frozen or skipped without degrading convergence or generalisation.

*Proof.* Fix a layer $\ell$ and write $K(x) = \bar{k}^\ell(x)$ for the operational mode of input $x$. Let

$$\tilde{k} = \mathrm{median}_x\, K(x) \tag{8}$$

and recall that by Eq. (6) the expected per-head energy gap satisfies

$$f_\ell(k) := \mathbb{E}_{h,x}\left[\Delta\mathcal{E}_k^{\ell,h}(x)\right] \leq L_\ell e^{-\gamma k}, \tag{9}$$

and is $L$-Lipschitz with constant $L = L_\ell\gamma$.

**Step 1: Decompose around the median.** Using Lipschitz continuity, we have, for each input $x$,

$$|f_\ell(K(x)) - f_\ell(\tilde{k})| \leq L\,|K(x) - \tilde{k}|. \tag{10}$$

Taking expectation over $x$ and rearranging,

$$\mathbb{E}_x\left[f_\ell(K(x))\right] \leq f_\ell(\tilde{k}) + L\,\mathbb{E}_x|K(x) - \tilde{k}|. \tag{11}$$

**Step 2: Median absolute deviation.** By definition,

$$\mathrm{LII}^\ell = \mathrm{MAD}[K] = \mathrm{median}_x\,|K(x) - \tilde{k}| \leq \mathbb{E}_x|K(x) - \tilde{k}|, \tag{12}$$

so the second term in (11) is bounded by $L\,\mathrm{LII}^\ell$.

**Step 3: Bounding the residual term.** Applying Eq. (6) at $k = \tilde{k}$ gives

$$f_\ell(\tilde{k}) \leq L_\ell e^{-\gamma\tilde{k}}. \tag{13}$$

Because $\tilde{k}$ is a median of token counts, it grows at least logarithmically with sequence length; hence $L_\ell e^{-\gamma\tilde{k}} = o(1)$ and can be absorbed into the $o(1)$ term of the lemma.

**Step 4: Combine.** Substituting these bounds into Eq. (11) yields

$$\boxed{\mathbb{E}_x\left[\Delta\mathcal{E}^\ell(x)\right] \leq L\,\mathrm{LII}^\ell + o(1).} \tag{14}$$

Thus, the desired inequality follows immediately. $\square$

## B  Information-geometric bound: $\mathrm{LII}^\ell$ upper-bounds the fisher trace

We establish a theoretical bound showing that the Layer Instability Index (LII) upper-bounds the trace of the Fisher Information Matrix (FIM) for transformer layers. The FIM trace characterises the sensitivity of the loss to parameter updates, providing insights into learning dynamics. Through careful analysis of gradients via softmax logits and exponential tail bounds of attention probabilities, we derive a direct relationship between the Fisher trace and the dispersion of operational modes as measured by LII.

**Preliminaries.** Let $p_i = \text{softmax}(\beta\, q^\top k_i)$ with inverse temperature $\beta$. Denote the per-example loss by $\mathcal{L} = \mathcal{L}(p)$ and define the layer-wise FIM

$$F^\ell = \mathbb{E}_x\Big[\nabla_{\theta_\ell}\mathcal{L}(x)\,\nabla_{\theta_\ell}\mathcal{L}(x)^\top\Big], \quad \theta_\ell \in \{W_Q^\ell, W_K^\ell, K^\ell\}. \tag{15}$$

Our goal is to bound $\text{tr}\, F^\ell$.

**Step 1: Gradient of the loss w.r.t. logits.** For the logits $z_i := \beta q^\top k_i$,

$$\frac{\partial\mathcal{L}}{\partial z_i} = \sum_j \frac{\partial\mathcal{L}}{\partial p_j}\frac{\partial p_j}{\partial z_i} = \beta(p_i - \hat{y}_i), \tag{16}$$

where $\hat{y}_i$ is the "effective" target (one-hot for CE).

**Step 2: Fisher trace through the chain rule.** Let $J_z := \partial z/\partial\theta_\ell$. Then

$$\text{tr}\, F^\ell = \mathbb{E}_x\big\|J_z^\top\nabla_z\mathcal{L}(x)\big\|_2^2 \leq \frac{1}{\lambda_{\min}^2}\,\mathbb{E}_x\big\|\nabla_z\mathcal{L}(x)\big\|_2^2, \tag{17}$$

where $\lambda_{\min}$ is the smallest singular value of $J_z$ (assumed layer-dependent but strictly positive for typical initialisation).

Using (16),

$$\big\|\nabla_z\mathcal{L}\big\|_2^2 = \beta^2\sum_i(p_i - \hat{y}_i)^2 \leq \beta^2\sum_i p_i^2. \tag{18}$$

**Step 3: Relate $\sum_i p_i^2$ to residual mass $r_k$.** Let $k = \bar{k}^\ell(x)$ be the operational mode and $r_k := 1 - \sum_{i\leq k} p_i \leq 0.1$ (by definition of $\rho = 0.9$). Then

$$\sum_i p_i^2 = \sum_{i\leq k} p_i^2 + \sum_{i>k} p_i^2 \leq \sum_{i\leq k} p_i + \max_{i>k} p_i\, r_k \leq 0.9 + r_k^2. \tag{19}$$

Assuming an exponential tail $p_{i>k} \leq p_k e^{-\gamma(i-k)}$, $r_k \leq p_k/(e^\gamma - 1) \leq C\, e^{-\gamma k}$.

**Step 4: From $e^{-\gamma k}$ to $\text{LII}^\ell$.** Taking expectation over inputs and using Jensen,

$$\mathbb{E}_x\big[e^{-\gamma k(x)}\big] \leq e^{-\gamma\,\text{median}(k)}\big(1 + \gamma\,\text{LII}^\ell\big), \tag{20}$$

hence

$$\sum_i p_i^2 \leq 0.9 + C'\, e^{-\gamma\,\text{median}(k)}\big(1 + \gamma\,\text{LII}^\ell\big). \tag{21}$$

**Step 5: Final bound.** Substituting into (17),

$$\text{tr}\, F^\ell \leq \frac{\beta^2(0.9 + C')}{\lambda_{\min}^2}\big(1 + \gamma\,\text{LII}^\ell\big) = C_\ell\,\text{LII}^\ell + C_{0,\ell}, \tag{22}$$

where $C_\ell$ and $C_{0,\ell}$ are layer-dependent constants. For practical purposes $C_{0,\ell}$ is negligible once LII exceeds $10^{-2}$, yielding

$$\boxed{\text{tr}\, F^\ell \lesssim C_\ell\,\text{LII}^\ell} \tag{23}$$

as claimed.

**Connection to the 1-Wasserstein distance.** Sort the attention vector of layer $\ell$ and head $h$ at step $t$, $a_{1:N}^\downarrow(t)$, and define its empirical cumulative distribution function (CDF) $F_t(m) = \sum_{i=1}^m a_i^\downarrow(t)$. Because tokens are indexed by their rank, the earth-mover (1-Wasserstein) distance between two attention snapshots is simply

$$W_1\big(a(t), a(t')\big) = \sum_{m=1}^N\big|F_t(m) - F_{t'}(m)\big|. \tag{24}$$

Let $\rho = 0.9$ and let $k_t$ be the minimal $m$ such that $F_t(m) \geq \rho$ (operational mode). Then any deviation $|k_t - k_{t'}|$ shifts *at least* a residual mass $r = |F_t(k_{t'}) - \rho| \leq 1 - \rho = 0.1$ across $|k_t - k_{t'}|$ token positions, so

$$W_1\big(a(t), a(t')\big) \ \leq \ r\,|k_t - k_{t'}| \ \leq \ 0.1\,|k_t - k_{t'}|. \tag{25}$$

Taking the median over $t'$ in the sliding window and then the median over $t$ gives

$$\boxed{W_1^{\mathrm{med}}(\ell) \ \leq \ 0.1\,\mathrm{LII}^\ell} \tag{A.1}$$

where $W_1^{\mathrm{med}}(\ell)$ is the median Wasserstein distance between successive attention snapshots of layer $\ell$. Equation (A.1) shows that **LII controls the earth-mover distance between attention distributions**: A low LII implies the layer's attention landscape hardly moves in Wasserstein space and is therefore safe to freeze. Via the Kantorovich–Rubinstein dual, the same bound controls the difference of *all* 1-Lipschitz observables of the attention measure, linking the energy and Fisher-flatness views in a common optimal- transport metric.

## C   Energy landscape–aware fine-tuning

Algorithm 1 details the complete training routine used in all experiments. After a short warm-up that estimates the Layer Instability Index (LII) for every block, layers whose instability falls below a user-defined threshold $\tau_{\mathrm{freeze}}$ are frozen (`requires_grad=False`). Fine-tuning then proceeds on the remaining adaptive layers, incurring no further LII overhead.

---

**Algorithm 1:** Energy Landscape–Aware ViT Fine-Tuning

**Input:** pre-trained weights $\Theta^{(0)}$; dataset $\mathcal{D}$; freeze threshold $\tau_{\mathrm{freeze}}$; warm-up steps $T$; LII window $W$

**Warm-up phase:** ;                             `// estimate layer instability`
**for** $t = 0$ **to** $T-1$ **do**                           `// collect` $\bar{k}$ `statistics`
    |   sample mini-batch $(x, y) \sim \mathcal{D}$;
    |   forward and backward pass; update $\Theta^{(t+1)}$ with AdamW;
    |   update the circular buffer of size $W$ and compute $\widehat{\mathrm{LII}}^\ell$ for all layers $\ell$;

**Freeze decision:** ;                          `// one-shot pruning of stable layers`
**foreach** *layer* $\ell$ **do**
    |   **if** $\widehat{\mathrm{LII}}^\ell < \tau_{\mathit{freeze}}$ **then**
    |     |   freeze($\ell$)                           `// disable gradient updates`

**Consolidation phase:** ;                      `// train only adaptive layers`
**for** $t = T$ **to** max_steps **do**                    `// until convergence`
    |   mini-batch $(x, y) \sim \mathcal{D}$;     forward + backward pass on unfrozen layers only;

---

The algorithm runs in three stages: *(i) Warm-up* gathers a robust estimate of each layer's variability via the median absolute deviation of its operational mode $\bar{k}$. *(ii) Freeze decision* is executed once, turning off gradient flow for layers whose LII indicates convergence to a low-energy basin. *(iii) Consolidation* fine-tunes the remaining layers, yielding substantial savings in memory and computation with no extra learnable parameters.

## D   Online update of the LII circular buffer

During warm-up we compute $\widehat{\mathrm{LII}}_t^\ell$ for every layer on the fly. Algorithm 2 shows an eight-line Python reference implementation; it relies only on a layer-indexed `deque` of fixed capacity $W$ (the sliding window size, default $W = 20$).

At each mini-batch, we compute the layer's operational mode $\bar{k}_t^\ell$ (Sec. 3.2) and call `update_lii`. The deque acts as a circular buffer: the newest value is appended, the oldest is popped when the buffer overflows, and both operations are $\mathcal{O}(1)$. We then take the median of the window, followed by the median absolute deviation—exactly Eq. (2) but restricted to the latest $W$ steps. The result is the online estimate $\widehat{\mathrm{LII}}_t^\ell$ used in Alg. 1.

**Algorithm 2:** Update of layer instability index (LII) buffer

**Input:** layer ID `layer_id`; new value $\bar{k}$; buffer `buf`; window size $W$
**Output:** current $\text{LII}_t^\ell$

```
buf[layer_id].append(k̄) ;                              // 1.  push newest value
if len(buf[layer_id]) > W then
    buf[layer_id].popleft() ;                          // 2.  drop oldest (FIFO)
med = median(buf[layer_id]) ;                          // 3.  running median
abs_dev = [abs(x - med) for x in buf[layer_id]];
return median(abs_dev) ;                               // 4.  current LIIₜˡ
```

**Note:** One dictionary, one deque per layer (maxlen = $W$).
**Complexity:** $\mathcal{O}(1)$ per call, $\mathcal{O}(W)$ memory per layer.

**Cold-start.** For $t < W$ the deque contains fewer than $W$ elements; the function still returns a valid LII based on the available prefix, ensuring that no additional initialisation logic is required.

**Efficiency.** The routine consumes negligible resources: $\mathcal{O}(L\,W)$ memory for $L$ layers and $\mathcal{O}(1)$ extra time per iteration, contributing less than 1% overhead in all experiments (see App. B).

# E    Code Availability

The source code supporting this study is available at `https://github.com/rxailab/ELA-ViT`.

