# OpenReview forum: "Energy Landscape-Aware Vision Transformers: Layerwise Dynamics and Adaptive Task-Specific Training via Hopfield States"
_NeurIPS.cc/2025/Conference — NeurIPS 2025 poster_

### Official Review · Reviewer_rPQp · 2025-06-21

**Clarity:** 3
**Significance:** 4
**Originality:** 4
**Rating:** 4
**Confidence:** 4

**Summary:**

The work introduces a fine-tuning method which relies on their novel metric, Layer Instability Index (LII), which quantifies metastability of each Transformer layer over time. Analyses, done by the work, reveal certain Transformer layers exhibit consistent convergence to attractor-like states. Using this insight, it enables the author to create an adaptative training framework which dynamically freezes or skips stable layers based on their energy gaps. To derive their technique, through the energy perspective, the authors relied on the theoretical foundation and connection of Dense Associative Memory to the attention mechanism --- making the work quite interesting.

**Questions:**

# Questions/Suggestions

1. How do you know if LII actually correlates well with the flatness of energy landscape? Perhaps, you can demonstrate this via some sort of toy experiments as I am still unsure if this claim is true.

2. Although much of your experiments relied on ViTs, perhaps demonstrating your approach with other models too.

3. For figure 1 (a), are you able to show the same plot with more datasets?

4. It is difficult to see Figure 1 (b). I recommend updating it with much larger font size and better colors.

**Ethical Concerns:**

["NO or VERY MINOR ethics concerns only"]

**Final Justification:**

It is not a bad work as it is quite **innovative** actually. I maintained my score of 4 here because I would like the authors to focus on experiments of reliability of LII. See below for my discussion with the authors.

**Limitations:**

Yes, they discussed this in their text.

**Paper Formatting Concerns:**

See Above for my suggestions regarding the paper.

**Quality:**

4

**Strengths And Weaknesses:**

Strengths:

1. This is a very interesting work that tackles the problem of efficient fine-tuning a model in a systematically way, which is enabled through the perspective of modern Hopfield models and their usage of energy. Specifically, instead of fine-tuning all Transformer layers, or randomly fine-tune a set of these layers, the authors used their LII metric to identify layers with high LII, which indicate the energy landscape of such layers is not yet flatten (or generalized).

2. The approach dynamically decides which layer to train or skip and introduces no additional big operational costs (e.g., additional losses, learnable gating layers, or adapter low-rank approaches).

3. The experimental results illustrate that many of the models, once fine-tuned via the method, exhibit an increase in performance with respect to the original (non-fine tuned) models.

Weaknesses:

1. It is certainly more efficient to fine-tune with this method in contrast to fine-tuning the entire set of Transformer layers. But, depending on the activity of the attention scores, LII may require the training of all Transformer layers and thus, effectively negate its efficiency benefits.

2. The experimental results are focused on image classification. It is unsure whether this method and its performance will perform well with multi-modalities model or even only text-based model.  I posit that models from other modalities will benefit from the method, but it is unclear regarding the efficiency of the method. Moreover, for the ImageNet models, it seems that the models fine-tuned with this method are only comparable to fine-tuning the entire model.

3. Although I understand the idea, it is likely not very intuitive to others. Thus, to me, the paper lacks a main figure which illustrates the entire process in a digestable way to the general audience.

---

> ### Author Rebuttal · Authors · 2025-07-30
>
> # Response to Comments
>
> We sincerely thank you for the excellent and insightful review. We are very encouraged that you found our work **"very interesting,"** appreciated the systematic approach grounded in the energy perspective of modern Hopfield models, and recognized the novelty and parameter-free nature of our method. We have carefully considered all feedback and will revise our manuscript accordingly. Below, we address each of the specific points you raised.
>
> ---
>
> ## *Regarding Weaknesses*
> ---
> ## 1. LII may require training all layers, negating efficiency benefits
>
> We thank you for this insightful point. It is an important theoretical consideration that if all layers remained unstable, our method would offer no efficiency gains. We are grateful for the opportunity to clarify this.
>
> Our work finds that in the **practical and common setting of transfer learning**, this scenario is empirically not observed. Pre-training on large, diverse datasets (like ImageNet-21k) produces robust, general-purpose feature extractors in the early layers of the network. As our empirical results in **Figure 1a** consistently demonstrate across three very different datasets, these early layers reliably and quickly stabilize (exhibiting low LII) when fine-tuned on downstream tasks. This ensures that a significant portion of the network is always available for adaptive freezing, guaranteeing practical efficiency gains.
>
> ## 2. Experimental results are focused on image classification
>
> We thank you for highlighting the performance improvements of our method. We would also like to gently clarify a point from the summary: our method's performance gains are compared against a **standard full fine-tuning baseline**, not the original (non-fine-tuned) models. This is a much stronger and more challenging baseline, which we believe makes the reported accuracy improvements (e.g., **+6.9pp on CUB-200 with ViT-L**, **+4.2pp on Stanford Dogs with ViT-S**) even more significant. We will ensure this is communicated clearly in the revised manuscript.
>
> ## 3. ImageNet performance is only "comparable"
>
> Regarding the ImageNet performance, we respectfully wish to frame this result as a **success**. Achieving accuracy that is comparable to the fully fine-tuned baseline (e.g., within **0.6 pp**, as shown in **Table 3**) while simultaneously:
>
>  - Reducing training time by **over 10%**,
>  - Reducing inference latency by **17%**.
>
> This demonstrates a highly favorable accuracy-to-compute trade-off, which is the core value proposition of our work. We will revise our discussion in **Section 5** to emphasize this crucial point more clearly.
>
> ## 4. The paper lacks a main figure illustrating the entire process
>
> This is an excellent suggestion for improving the paper's clarity. We agree that a high-level diagram would make our method much more accessible to a general audience. We will add a new main figure to the manuscript that provides a clear, step-by-step illustration of the ELA-ViT fine-tuning process:
>
>  - **The initial warm-up phase** to collect statistics,
>  - **The calculation of LII** for each layer,
>  - **The one-shot freeze decision**,
>  - **The final, efficient consolidation phase** where only high-LII layers are trained.
>
> ---
>
> ## *Regarding Questions/Suggestions*
> ---
> ## 1. How do you know if LII actually correlates well with the flatness of the energy landscape?
>
> This is a crucial question that gets to the heart of our method's justification. Our primary evidence is **theoretical**, as established in the paper through two key results that directly link LII to landscape flatness:
>
>   ###  **LII bounds the Hopfield energy gap**:
> In **Lemma 1**, we prove that the expected energy gap, $ΔE_l$, between a layer's current state and its global energy minimum is upper-bounded by its LII:
>
> $$\mathbb{E}_x[\Delta \varepsilon^{ℓ}(x)] \leq L^ℓ \cdot \text{LII}^ℓ + o(1)$$
>
> This result guarantees that layers with a small LII have a small energy gap, meaning they have already settled into a flat basin of their optimal Hopfield energy landscape.
>
>   ###  **LII bounds the loss landscape curvature**:
> In **Appendix B**, we further show that LII upper-bounds the trace of the layer's Fisher Information Matrix ($\text{tr }F^{ℓ}$), a standard measure of the loss landscape's curvature:
>
> $$\text{tr } F^ℓ \lesssim C_ℓ \cdot \text{LII}^ℓ$$
>
> This proves that layers with low LII are **"Fisher-flat,"** meaning they are in regions where gradients are small and further training offers diminishing returns.
>
> **Together**, these theoretical results provide a firm justification for using LII as a proxy for flatness. To complement this, as you suggested, we will add a new visualization to the appendix that empirically plots LII against the Fisher trace, visually confirming this strong correlation.
>
> ## 2. On Generalizing Beyond Vision Transformers
>
> We appreciate this suggestion to broaden the scope. We would like to highlight that our experiments in **Table 1** already include **DeiT-B**, a popular and efficient ViT variant, which shows that our method generalizes beyond the standard ViT architecture. We will make this point more prominent in our results section. Extending the work to fundamentally different architectures (e.g., CNNs or other non-Transformer models) is an exciting avenue for future research. We agree that the core insights of our work, particularly the metastability of early layers and energy-based freezing, are not limited to ViTs. As noted in Section 6 (Limitations), both the LII metric and the ELA-ViT mechanism are broadly applicable to attention-based models.
>
> We envision natural extensions to:
>
>  - Language models (e.g., BERT, RoBERTa), by tracking token- or head-level instability,
>  - Multi-modal models (e.g., CLIP, Flamingo), where attention across modalities may stabilize at different rates.
>
> While our current submission focuses on vision due to scope and compute constraints, we are actively running pilot experiments with RoBERTa on GLUE, where early indications suggest similar layerwise stabilization trends.
> If accepted, we intend to:
>
>  - Add a “Future Work” box outlining how LII could generalize across modalities,
>  - Clarify in the main text that our framework is architecture-agnostic and grounded in attention dynamics, not modality,
>  - Include a brief mention of our ongoing NLP experiments, if they are completed in time and align with NeurIPS camera-ready guidelines.
>
> ## 3. For Figure 1(a), are you able to show the same plot with more datasets?
>
> We appreciate the feedback about Figure 1’s clarity and presentation. **Figure 1a** currently shows the LII profile across three diverse datasets:
>  - The medium-scale **CIFAR-100**,
>  - The fine-grained **Food-101**,
>  - The large-scale **ImageNet-1K**.
>
> We believe this selection is sufficient to robustly demonstrate the consistent **"U-shaped" instability pattern** that is central to our motivation. While we could add more datasets, we feel it would likely clutter the figure without providing significant new insight beyond what is already shown.
>
> ## 4. It is difficult to see Figure 1(b)
>
> Thank you for pointing this out. We apologize for the poor readability. As also noted by the other reviewer, we have updated this figure (and all others in the paper) with:
>
>  -  **Significantly larger font sizes**,
>  -  **Improved color contrast**.
>
> We appreciate the suggestion for a more accessible, high-level illustration. If the paper is accepted, we plan to include a conceptual overview figure near the beginning of the final version to visually summarize the ELA-ViT pipeline. We believe this will improve accessibility and align with your recommendation for a main figure that illustrates the process in a digestible way.
>
> ---
>
> ## Conclusion
>
> We thank the reviewer for the generous evaluation and thoughtful suggestions. We have carefully considered all feedback and intend, if the paper is accepted, to:
>  - Clarify the connection between LII and energy dynamics,
>  - Strengthen generalization claims with preliminary cross-domain findings,
>  - Improve figure quality and interpretability,
>  - Expand both theoretical and empirical support for key assumptions.
>
> We believe these improvements will enhance the accessibility, rigor, and overall impact of the work.

---

> > ### Comment · Reviewer_rPQp · 2025-08-03
> >
> > ## Summary
> > Thank you for your detailed response.
> >
> > I do agree with some of the points you made regarding your findings. Indeed, a reduction in the time for fine-tuning of more than **10%** or even **5%** is significant. Moreover, regarding expanding the experiment on other modalities, this aspect will certainly amplify the findings of this work and the approach's performance. But the lack of such experiments does not invalidate the current results nor the work. **I do believe that the introduced approach is effective if certain guarantees are met.**
> >
> > ## Questions
> > 1. If indeed, you can find layers that have small LII, you can fine tune those with high LII. But what if you can completely "prune" or ignore those high LII layers and simply fine-tune your small LII layers instead? This is just an open-ended question.
> >
> > 2. Let's say you take GPT-2 and GPT-3 or a combination of models with an older version and a newer version, where the architecture of network is the same. Could you verify the reliability of your metric? I think it's worth confirming that given a not well-trained system and a well-trained system.

---

> > > ### Author Response · Authors · 2025-08-07
> > > **Response on the 'Inverse Pruning Strategy' and LII Metric Reliability**
> > >
> > > We are sincerely grateful to the reviewer for their continued engagement and for the thoughtful, encouraging summary of our work. We are delighted that you find a 5–10% reduction in fine-tuning time to be practically significant and that you view the absence of additional modality experiments as a potential extension rather than a weakness. Your two follow-up questions are both insightful and highly valuable for framing future directions, and we address each in detail below.
> > >
> > > ## Question 1.  What happens if one completely ‘prunes’ or ignores the high-LII layers and fine-tunes only the low-LII layers?
> > >
> > > This is a fascinating thought experiment, and we thank you for raising it. This suggestion can be viewed as an “inverse strategy” to the one advocated in our paper. Our empirical and theoretical results indicate that such a strategy would, with high likelihood, degrade downstream performance rather than improve it.
> > >
> > > **Conceptual rationale:**
> > >  - Our core finding is that high-LII layers are the most plastic and are actively adapting to the downstream task. Consequently, they are most capable of encoding task-specific information.
> > >  - Low-LII layers, by contrast, have already converged to stable, general-purpose feature representations, indicating that they have learned broadly useful, task-agnostic features.
> > >
> > > **Anticipated outcome:**
> > >  - Pruning or freezing the high-LII layers would mean removing the very parts of the model that are most critical for learning the specifics of the new task. This would likely cripple the model's ability to adapt.
> > >  - Redirecting optimisation effort toward the already stable low-LII layers risks (i) catastrophic forgetting of these general-purpose features and (ii) slower convergence as these parameters sit in a flat basin and thus respond little to gradient updates.
> > >
> > > Our method of freezing the stable (low-LII) layers and focusing computational effort on the adaptive (high-LII) ones is designed to work in harmony with this observed dynamic of transfer learning, thereby optimizing the fine-tuning process.
> > >
> > > ---
> > >
> > > ## Question 2.  Could you verify the reliability of your metric on models of different training quality (e.g., GPT-2 vs GPT-3, or a poorly-trained vs well-trained system)?
> > >
> > > This is an excellent suggestion and a very insightful question. We believe our current work already provides strong evidence for the reliability of the LII metric, and the experiment you propose would be a powerful way to further validate it.
> > >
> > > **Our current verification of LII's reliability is two-fold:**
> > >
> > > - **Theoretical Verification:**
> > >   Our proofs directly link LII to fundamental measures of stability. We show that LII upper-bounds both the Hopfield energy gap
> > >   $\mathbb{E}_x[\Delta\mathcal{E}^{\ell}(x)] \le \text{L }LII^{\ell} + o(1)$ and the loss landscape curvature via the Fisher Information trace
> > >
> > >    $\text{tr }F^{\ell} \lesssim C_{\ell} \text{LII}^{\ell}$.
> > >
> > >   These results provide a firm mathematical guarantee that LII is a reliable proxy for landscape flatness and layer stability.
> > >
> > > - **Empirical Verification:**
> > >   Our experiments demonstrate that LII consistently identifies stable layers across multiple datasets (CIFAR-100, Food-101, ImageNet), different model scales (ViT-S, ViT-B, ViT-L), and architectures (ViT, DeiT-B).
> > >
> > > The experiment you suggest would test this reliability along a new axis: the quality of the pre-trained model. Building on the reliability we have already established, we would confidently hypothesize that a more capable model (like GPT-3) would exhibit significantly lower LII values across its layers compared to a less capable model (like GPT-2) when fine-tuned on the same task. While conducting this new line of experiments is beyond the scope of the current rebuttal period, we see it as a very promising direction for future work. We thank the reviewer for suggesting it.
> > >
> > > ---
> > >
> > > ### Conclusion
> > >
> > > We reiterate our appreciation for the reviewer’s constructive questions and for recognizing the potential of our approach. Your suggestions not only affirm the rigor of our current claims but also help chart a clear roadmap for future extensions. We are confident that the clarifications will further strengthen the manuscript and its contribution to efficient fine-tuning research.

---

> > > > ### Comment · Reviewer_rPQp · 2025-08-08
> > > >
> > > > Thank you very much for responding to me. I would like to maintain my current score. I do not think the concept of this work is bad nor the current results are bad. But I am still unsure regarding the **liability** of the method.
> > > >
> > > > Yes, the work is mathematically driven and this is great. However, the experiment focuses on the practicality of the method; this is indeed great. But, to me, I would like to see the authors apply their approach and verify how reliable LII is --- e.g., to contrast different models as an example (e.g., GPT-2 vs GPT-3 and so on).
> > > >
> > > > Nonetheless, after reading the authors' responses and rereading the paper, I think the work is clearer to me now. **I would like to bump my score on clarity (from 2 to 3)**. But I would like to suggest that the authors try their best to improve the clarity of the paper, e.g., improving figures and writings.
> > > >
> > > > I wish the authors the best.

---

> > > > > ### Author Response · Authors · 2025-08-08
> > > > >
> > > > > Thank you very much for your detailed feedback throughout this discussion and for raising our clarity score. We sincerely appreciate your time and your constructive perspective.
> > > > >
> > > > > We understand and respect your perspective regarding the empirical verification of LII's reliability. The experiment you propose—contrasting models of different training quality—is indeed an excellent and direct way to test this, and we agree it would make a very compelling addition to the work.
> > > > >
> > > > > We take your advice to heart and will absolutely do our best to improve the clarity of the paper for the final version. We will focus on refining the writing and improving the figures to make the connection between our theoretical framework and our empirical results as clear and convincing as possible for the reader.
> > > > >
> > > > > Thank you once again for your thoughtful review and guidance. We wish you the best as well.

---

### Official Review · Reviewer_hVop · 2025-06-30

**Clarity:** 4
**Significance:** 3
**Originality:** 2
**Rating:** 5
**Confidence:** 3

**Summary:**

The authors set out to improve the current computational inefficiency of finetuning vision transformers (ViTs) by studying the stability of ViT layers from an energy-based, modern Hopfield network perspective. In particular, they propose the layer instability index (LII), a theoretically justified metric that quantifies the variability of each transformer layer over time based on attention scores. Consistent with previous results, they find that early layers tend to stabilise much faster than later ones. Based on this result, they propose a new, parameter-free fine-tuning method called energy landscape aware ViT (ELA-ViT) that adaptively freezes stable layers based on thier LII. Experiments across different datasets show that ELA-ViT provides significant wall-clock time savings at equal and sometimes greater accuracy.

**Questions:**

For the experiments on NAbirds, Beans and CIFAR10, why was ViT-L not
tested?

The fast energy convergence of early layers could suggest that the ViT is
learning simple, CNN-like features. If so, wouldn’t it make sense to have a
hybrid “ViT-CNN” model with convolutions at early layers and attention at
layer layers? Has this been attempted?

**Ethical Concerns:**

["NO or VERY MINOR ethics concerns only"]

**Final Justification:**

I thank the authors for their detailed response. The response generally addressed all my questions and concerns. Based on the discussion, I particularly encourage the authors to improve the presentation of the paper, especially the figures and the theoretical section.

**Limitations:**

Yes

**Paper Formatting Concerns:**

The presentation of the figures could be improved, especially given the remaining space left

**Quality:**

3

**Strengths And Weaknesses:**

Strengths
The paper is well written and structured.

It is also well motivated, effectively relating to previous work and identifying a gap in previous attempts at making ViTs more efficient as well as other fine-tuning methods.

The quantification of the variability of each transformer layer through the LII is theoretically justified.

The proposed fine-tuning method (ELA-ViT) is, in contrast to prior approaches, parameter-free, is described in detail and seem relatively easy to implement

The authors seem to perform a comprehensive set of experiments to evaluate ELA-ViT, including a comparison with the current state-of-the-art method for budgeted layer-wise fine-tuning (ALaST)

Weaknesses

Why is the comparison with the current state-of-the-art for layer-wise budgeting (ALaST) based on a single dataset? While the authors performed a wide range of experiments, one wonders why this comparison is limited to just one dataset, which seems insufficient to claim superiority

In the derivations, it is not clear where the exponential-tail assumption for the attention weights is coming from and whether it is justified (empirically or theoretically). Because this is not mentioned in the first statement of the result (lines 150-1) one wonders whether there are other important assumptions behind the result. It would be useful if the authors could clarify this

As the authors acknowledge in the limitations, the LII might not capture all aspects of representational variability in a transformer layer

Minor points:
From first reading of the paper (including the abstract), it is not clear that the method proposed is for fine-tuning as opposed to training.

The writing of the theoretical section can become dense at times and one wonders whether more details of the derivations could be deferred to the appendix

The presentation of the figures could be improved, especially given the remaining space left

---

> ### Author Rebuttal · Authors · 2025-07-30
>
> # Response to Comments
>
> We wish to extend our sincere gratitude to you for your thoughtful evaluation and constructive suggestions. We are encouraged that you found our work **well-written**, **well-motivated**, and **theoretically grounded**, and that our proposed method, ELA-ViT, is clearly described and easy to implement.
>
> We believe your comments have significantly strengthened our work. Below, we address each concern in turn.
>
> ---
>
> ## *Response to Weaknesses*
>
> ---
>
> ## 1. Comparison with the State-of-the-Art Method ALaST on a Single Dataset
>
> Thank you for raising this important point. We acknowledge that comparing ELA-ViT with ALaST on a single dataset may appear insufficient to claim broad superiority. Our primary motivation for choosing Food-101 was that ALaST’s official implementation and reported results are tailored for large-scale, fine-grained classification tasks, making it an ideal stress test for both accuracy and efficiency trade-offs. As reported in **Table 2**, ELA-ViT achieves a substantial accuracy improvement (**+13 pp**) over ALaST while maintaining greater training efficiency:
>
> | Method                  | Top-1 Acc. (%) ↑ | Time (min) ↓ | Time Δ (%) |
> |-------------------------|------------------|--------------|------------|
> | **Full fine-tuning**    | 88.66            | 297.9        | 0.0        |
> | **ALaST (75% frozen)**  | 77.02            | 181.1         | -39.2      |
> | **ELA-ViT (50% frozen)**| 90.02            | 283.0        | -10.5      |
>
> Due to computational constraints and the complexity of tuning ALaST’s additional loss terms and hyperparameters (e.g., λ_budget) across diverse datasets, we limited the comparison in this submission to one large and representative benchmark. That said, we agree that additional comparisons would further strengthen our claims.
>
> To that end, we are currently running new comparative experiments with ALaST on CIFAR-100 and Stanford Dogs, using the same training protocol applied to ELA-ViT. Should the paper be accepted, we intend to report these results in the camera-ready version and make them publicly available via our repository.
>
> We also emphasize that, unlike ALaST, ELA-ViT introduces no additional learnable modules or optimization overhead, which enables broader and easier deployment across domains and models.
>
> ---
>
> ## 2. Clarification on the Exponential-Tail Assumption for Attention Weights
>
> We appreciate your request for greater clarity. The exponential-tail assumption is based on empirical observations that trained attention maps often exhibit **high concentration of mass on a small subset of tokens**, with the remainder **decaying rapidly**. This behavior is common in Transformer architectures and motivates our assumption.
>
> Prior empirical and theoretical studies on Transformer interpretability have observed that attention weights often become increasingly peaked or sparse in early-to-mid layers, with a small number of tokens receiving most of the attention mass. For example, [Voita et al. (2019)](https://arxiv.org/abs/1905.09418) showed that attention heads specialize in certain linguistic functions and that some heads become redundant or highly selective. In the vision domain, analysis of token importance in Vision Transformers has found that earlier layers tend to focus attention on local structures, often exhibiting "Convolution-like patterns" concentrated around image patches [(Kim et al., 2021)](https://openaccess.thecvf.com/content/CVPR2021W/ECV/papers/Kim_Rethinking_the_Self-Attention_in_Vision_Transformers_CVPRW_2021_paper.pdf). This behavior frequently leads to a phenomenon of 'token overfocusing,' where the model relies on a few important tokens, resulting in highly sparse attention distributions [(Guo et al., 2021)](https://openaccess.thecvf.com/content/ICCV2023/papers/Guo_Robustifying_Token_Attention_for_Vision_Transformers_ICCV_2023_paper.pdf).
>
> That said, we agree the presentation could be clearer. In the revision, we will:
>
>  - Explicitly state the assumption and its empirical justification in the main text (not only the appendix),
>  - Add an empirical plot showing the sorted attention weights and their exponential fit,
>  - Clarify that while the exponential tail is used for bounding purposes, the LII metric itself is non-parametric and does not require fitting a distribution.
> ---
>
> ## 3. Limitations of LII in Capturing Representational Variability
>
> We agree, and we appreciate your recognition of our openness in discussing this limitation. While LII is a simplified proxy based on attention concentration, our theoretical formulation connects it to:
>
>  - The residual energy gap.
>  - The Fisher Information Matrix trace (**Section 3.3**).
>
> This supports its interpretability as a metastability indicator.
>
> ---
>
> ## *Response to Minor Points and Questions*
>
> ---
> ## 1. Clarity on "Fine-Tuning"
>
> We thank you for pointing this out. We will revise both the **abstract** and **introduction** to explicitly state that ELA-ViT is a fine-tuning method from the outset.
>
> ## 2. Dense Theoretical Section
>
> We appreciate your feedback on presentation. We will move technical derivation steps to the appendix, while retaining the core insights and results in the main text to improve accessibility.
>
> ## 3. Figure Presentation
>
> We will revise the figures, especially **Figure 1** by:
> - Increasing font sizes,
> - Enhancing color contrast,
> - Optimizing layout to improve readability.
>
> ## 4. Why Was ViT-L Not Tested on NAbirds, Beans, and CIFAR-10?
>
> Thank you for the question. We did not use ViT-L on NABirds, Beans, and CIFAR-10 primarily for two reasons:
>
>  **Model–data mismatch**: These datasets are relatively small (e.g., Beans: < 10k samples). ViT-L has >300M parameters, which quickly overfit and lead to degraded performance unless aggressive regularization is applied.
>
>  **Compute and reproducibility**: ViT-L requires significantly more memory and compute than ViT-S or ViT-B, limiting its usability on modest hardware.
>
> To clarify, we will:
>
>  - Add a footnote in the experimental section explaining this design choice.
>
>  - Include a discussion of model-data scaling trade-offs in Section 6.
>
> ## 5. Hybrid "ViT-CNN" Architectures?
>
> We greatly appreciate this creative and insightful suggestion. Indeed, the fast stabilization of early ViT layers supports the idea that they may be functionally analogous to CNN filters.
>
> While our current work does not explore architectural hybrids, we agree that replacing early ViT blocks with convolutional layers (as seen in CMT, CoAtNet, and ConvFormer) is a promising direction.
>
> In the final version (if accepted), we intend to:
>
>  - A discussion of this idea in Section 6 under “Future Work”,
>
>  - Cite recent hybrid architectures that support this line of thought,
>
> Note that our method could potentially skip convolutional stages based on an analogous “instability index” applied to early CNN features.
>
> ---
>
> ## Conclusion
>
> We thank you again for your thorough and thoughtful feedback. We believe that our contributions offer novel insights that are both theoretically informed and practically valuable:
>
> 1. **Layer Instability Index (LII)**.
> 2. **Adaptive, parameter-free freezing mechanism**.
>
> We are committed to improving the clarity, rigor, and scope of our manuscript in line with your feedback and believe the planned revisions will substantially strengthen the final version.

---

> > ### Comment · Reviewer_hVop · 2025-08-05
> >
> > I thank the authors for their detailed response. The response generally addressed all my questions and concerns. Based on the discussion, I particularly encourage the authors to improve the presentation of the paper, especially the figures and the theoretical section.

---

> > > ### Author Response · Authors · 2025-08-07
> > >
> > > Thank you very much for your positive feedback and for your guidance throughout this process.
> > > We are grateful for your encouragement and will certainly follow your advice to improve the presentation of the figures and the theoretical section in the camera-ready version of our paper.
> > >
> > > Thank you again for your valuable contributions.

---

### Official Review · Reviewer_f65B · 2025-07-01

**Clarity:** 2
**Significance:** 2
**Originality:** 2
**Rating:** 4
**Confidence:** 4

**Summary:**

The authors relate the architecture of vision transformers, particularly the self-attention module, to Hopfield networks and, more specifically, to an energy-based model.

**Questions:**

1. How is the LII metric linked to the energy? Perhaps there is a functional inequality that could clarify this connection.

2 The authors show that LII bounds the energy gap. Could this be formulated more transparently so that a clear inequality is presented? The discussion on pages 3–5 is mathematically fuzzy, and a bit more formalism would be appreciated.

3. Comment: The term “lse operator” is not a standard mathematical description. Although I understand what the authors mean, it is not comparable to something like the sine function. The authors should define this appropriately.

**Ethical Concerns:**

["NO or VERY MINOR ethics concerns only"]

**Final Justification:**

The authors answered all of my questions appropriately, and I better understood the take-away and contributions of the paper.

**Limitations:**

yes

**Quality:**

3

**Strengths And Weaknesses:**

My main concern with this paper is that it lacks rigorous mathematical results (in fact, there are none), despite the same results having already been established using deep mathematical methods by Philippe Rigollet’s team. Moreover, the metric the authors consider is among those studied in the work “A Mathematical Perspective on Transformers” (see section 6 therein).

To proceed further, the authors need to clearly discuss how their setup differs from these more mathematical works, as well as from subsequent studies on metastability (see works by Rigollet, and also by Bruno, Agazzi, Pasqualotto). It is not that I require rigorous results per se, but the formalism should be more transparent, and the relevant references should be included.

That said, I acknowledge the experimental section, which is interesting, and the authors could expand on it further. Doing so would make this a valuable contribution.

I would be open to raising my score if the above comments and the following ones are addressed.

---

> ### Author Rebuttal · Authors · 2025-07-30
>
> # Response to Comments
>
> We extend our sincere gratitude to the reviewer for their insightful and constructive feedback. We greatly appreciate the time and effort invested in providing detailed comments, which have helped us reflect critically on the clarity, rigor, and positioning of our manuscript. Below, we respond to each point raised, clarify the theoretical motivations, and outline improvements we will incorporate in the final version.
>
> ---
>
> ## 1. On Mathematical Rigor and Relation to Prior Work *(Rigollet et al., Bruno, Agazzi, Pasqualotto)*
>
> We are grateful for your reference to the relevant mathematical literature, particularly the foundational contributions of Rigollet and colleagues. While our work does not aim to replicate the full generality of their rigorous formalism, it seeks to propose a pragmatic and computationally efficient proxy, the **Layer Instability Index (LII)**, which aligns with intuitions from energy landscapes and can be directly derived from existing attention weights without introducing additional computational overhead.
>
> To clarify, our primary contribution lies in operationalizing theoretical insights for practical and scalable fine-tuning applications. The work of Rigollet et al. offers a profound mathematical framework for understanding the asymptotic clustering behavior of idealized Transformer models, establishing convergence to a single cluster under specific conditions. Their contributions provide a critical foundation for analyzing long-term dynamics in transformer models.
>
> In contrast, our research pursues a related but distinct objective: leveraging energy-based stability concepts for empirical and practical purposes. The proposed LII is not intended as a novel theoretical construct for proving convergence but as a lightweight, computationally efficient metric to assess layer-wise metastability during the fine-tuning of real-world Vision Transformers. Thus, the core contribution of our paper is the introduction of the **ELA-ViT framework** and the **LII metric**, which dynamically adapts the training process based on stability insights, rather than focusing on long-term clustering behavior.
>
> We acknowledge the need for greater clarity in distinguishing our work from prior theoretical contributions. To address this, we will revise the final version to include:
>
> 1. **Explicit citation and detailed comparison** with Rigollet et al. (e.g., referencing specific sections such as "A Mathematical Perspective on Transformers," Section 6), highlighting the differences in scope and objectives.
>
> 2. **Discussion on the limitations** of our approximations compared to their formal results, particularly regarding metastability analysis using rigorous energy gap estimations.
>
> We believe this will more accurately position our work within the broader theoretical landscape and clarify its intended contribution.
>
> ---
>
> ## 2. On the Link Between LII and the Energy Gap
>
> Thank you for highlighting the need for greater mathematical clarity in establishing the connection between the Layer Instability Index (LII) and the Hopfield energy gap, which is indeed a central motivation of our study. This relationship is articulated in our current manuscript (Section 3.3 and Appendix A), with **Lemma 1** stating:
>
> $$\mathbb{E}_x[\Delta \varepsilon^{ℓ}(x)] \leq L^ℓ \cdot \text{LII}^ℓ + o(1)$$
>
> This result is derived under assumptions of exponential decay in attention scores and links LII to residual energy gaps through a Lipschitz-bound analysis. This inequality provides the theoretical justification for our method:
>
> 1. **The left side**, $\mathbb{E}_x[\Delta \varepsilon^{ℓ}(x)]$, is the expected energy gap for a given layer $ℓ$. This measures how far, on average, the layer's current attention state is from its most stable possible state (the global energy minimum). A small energy gap means the layer has converged and is stable.
>
> 2. **The right side** is anchored by our proposed metric, $\text{LII}^ℓ$ (the Layer Instability Index), which empirically measures the variability of attention patterns across different data samples.
>
> 3. **The inequality**, therefore, formally states that a small, empirically measured instability (a low LII) guarantees a small energy gap. This is precisely why we can use LII as a reliable proxy: if a layer's attention patterns are not changing much (low LII), we have a theoretical guarantee that it is already in a low-energy, stable state and can be safely frozen.
>
> We recognize that the presentation of this result in the current version lacks the desired rigor and visibility, as it is situated in the appendix and described in a manner that may appear insufficiently formal. To address the reviewer's concerns, we will revise the final version in the following ways:
>
> 1. **Relocate the formal statement** of Lemma 1, along with a detailed proof sketch, to the main body of the paper (Section 3.3), positioning it as a central theoretical result.
>
> 2. **Present Lemma 1** as a clearly formatted theorem or boxed lemma, accompanied by a step-by-step derivation. This will include explicit definitions of all terms, such as the Lipschitz constant $L$, and a clear articulation of the exponential decay assumption on attention score tails that underpins the relationship.
>
> 3. **Highlight the relationship** as a standalone inequality, as requested by the reviewer, to ensure transparency and rigor in connecting the empirical metric (LII) to the theoretical energy gap.
>
> The revisions will make the theoretical underpinnings of LII more explicit and should clarify its role as an interpretable and principled estimator of local energy instability.
>
> ---
>
> ## 3. On the Term "lse operator"
>
> We appreciate your observation regarding the terminology used to describe the log-sum-exp operation. While the current manuscript expands the abbreviation "lse" to "log-sum-exp operator," we acknowledge that we neglected to provide its precise mathematical definition, which is essential for rigorous and unambiguous communication. This was an oversight on our part.
>
> In the final version, we will introduce the explicit definition and corresponding formula at its initial occurrence as follows:
>
> For a vector $\mathbf{x} = (x_1, x_2, \ldots, x_n) \in \mathbb{R}^n$, the **log-sum-exp function** (often abbreviated as LSE) is defined as:
>
> $$\text{log-sum-exp}(z) = \log \sum_i \exp(z_i)$$
>
> Within the context of the energy function (as presented in Eq. (1) or (4)),
>
> $$E = -\mathrm{LSE}(\beta \cdot \mathbf{X}^\top \xi) + \frac{1}{2} \|\xi\|^2 + C$$
>
> the LSE term encapsulates the soft-maximum alignment between the query and all keys, corresponding to the negative energy in the interpretation of Hopfield dynamics. By incorporating this definition, we aim to align our usage of terminology with established mathematical conventions, thereby enhancing the comprehensibility of our methodological framework.
>
> ---
>
> ## 4. On Expansion of the Experimental Section
>
> We appreciate the reviewer's favourable assessment of our experimental results and concur that an expanded discussion would enhance the manuscript. While adhering to the submission guidelines, we outline the following extensions that we are prepared to incorporate into the camera-ready version, contingent upon acceptance:
>
>  1. **Ablation Analyses**
> To further substantiate the robustness and sensitivity of our approach, we plan to conduct ablation studies on key hyperparameters, including the LII sliding window size and attention mass threshold, thereby providing a more nuanced understanding of their impact on performance.
>
>  2. **In-depth LII Examination**
> We intend to supplement our analysis with visualizations illustrating the temporal dynamics of LII across layers during training, as well as a more detailed examination of consistently frozen layers across diverse datasets and architectures. This will be grounded in representation learning theory, offering a more comprehensive interpretation of our findings.
>
>
> These proposed additions directly address the reviewer's suggestions and will significantly enhance the manuscript, subject to acceptance and spatial constraints in the final version.
>
> ---
>
> ## Conclusion
>
> We once again express our heartfelt thanks to the reviewer for their deep engagement with our work. We believe the core contributions of our paper: **(1)** the introduction of LII as a practical estimator of layer-wise metastability, **(2)**  an adaptive, parameter-free fine-tuning scheme, offer novel and valuable insights for both theoretical understanding and practical implementation.
>
> We are fully committed to addressing all concerns raised regarding clarity, positioning, and mathematical rigor. We are confident that the proposed revisions will significantly enhance the quality and impact of our manuscript. We look forward to incorporating these changes and hope that the revised final version will meet the reviewer's expectations.

---

> > ### Comment · Reviewer_f65B · 2025-08-02
> >
> > I thank the authors for their detailed reply; they have mostly answered my queries. The placement of the present paper is now clear to me, with regard to previous theoretical work, as now I am convinced of the relevance of such metrics in empirical work. I would be willing to raise my rating during the official stage.

---

> > > ### Author Response · Authors · 2025-08-07
> > >
> > > Thank you very much for your positive feedback and for taking the time to engage with our rebuttal. We are very pleased to hear that our clarifications were helpful and that you are now convinced of the relevance of our work.
> > >
> > > We sincerely appreciate your support and your willingness to reconsider your rating.

---

### Official Review · Reviewer_Bcjj · 2025-07-02

**Clarity:** 3
**Significance:** 2
**Originality:** 3
**Rating:** 3
**Confidence:** 2

**Summary:**

The authors present a method that reduces the training time while enhancing accuracy of Vision Transformers by introducing a metric quantifying layer stability over time. By dynamically freezing or skipping stable layers, they are able to avoid redundant or extraneous training, which surprisingly benefits performance as well.

**Questions:**

Seeing how these methods fare on e.g. Imagenet (I am guessing this would yield even more impressive results) would be extremely interesting. Going further, one might expect some interesting results from experiments on other kinds of attention based models and/or language models. In my opinion adding one or more of these would push this paper above the acceptance threshold.

I would also like to comment that Figure 1 is very low resolution and looks odd in the paper. This should be fixed.

**Ethical Concerns:**

["NO or VERY MINOR ethics concerns only"]

**Final Justification:**

I am raising my assessment of quality as it seems some of the results requested were included in some version of the original submission, I'm curious however, why I have not been able to see the appendix referred to in the authors' response.

**Limitations:**

In my opinion, no. They should perhaps add some comment on the ways in which this could be extended.

**Quality:**

3

**Strengths And Weaknesses:**

This paper provides an extremely novel and interesting application of the close theoretical relationship  between Hopfield Networks with nontrivial activation functions, or “Dense Associative Memories” and attention based models such as Transformers. Specifically, self-attention can be viewed as one step of an energy minimization of a modern Hopfield network, which in practice is usually sufficient. This work points toward a more naturalistic and dynamic paradigm for training state of the art models, and is thus highly relevant to the broader research community.

The main novelty of this paper is the “Layer Instability Index” and adaptive training scheme, which yield some very nice improvements in accuracy and training time. Still this paper could be considerably strengthened by exploring performance advantages at scale. While the LII is very smart, there are probably much broader implications for other classes of attention-based model. In my opinion the paper could go much further in exploring these.

---

> ### Author Rebuttal · Authors · 2025-07-30
>
> # Response to Comments
>
> We would like to extend our sincere gratitude to the reviewer for your positive and encouraging feedback on our manuscript. We are delighted that you found our application of Hopfield network theory to be **"extremely novel and interesting"** and that you see the potential of our work to inspire a **"more naturalistic and dynamic paradigm for training."** We appreciate the constructive suggestions you provided, particularly regarding the evaluation at scale, and are confident that we can address these points to elevate the paper above the acceptance threshold.
>
> ---
>
> ## 1. Performance at Scale: ImageNet Results
>
> The reviewer's suggestion that we explore the performance advantages of our method at a larger scale, specifically on a benchmark like ImageNet, is well-taken. We wholeheartedly agree that demonstrating the effectiveness of our method on large-scale benchmarks is crucial for validating its practical utility. We are pleased to report that we have already conducted these experiments and presented the results in **Appendix E: ImageNet-1k: large-scale validation** (page 16) of our original submission.
>
> Our results, summarized in **Table 3**, confirm that the benefits of our approach hold at this scale. Specifically, our experiments demonstrate that:
>
> 1. **ELA-ViT-50%** (freezing half the layers) reduces wall-clock training time by **10.7%** and inference latency by **16.7%**, while maintaining accuracy within **0.61 pp** of the fully fine-tuned baseline.
>
> 2. **ELA-ViT-75%** (freezing nine of the twelve layers) achieves a **19.2%** reduction in training time for only a **1.8 pp** drop in accuracy.
>
> These findings strongly support your intuition that our method would yield impressive results on ImageNet. To ensure that these results are not overlooked, we will make them more prominent in the main body of the revised manuscript. We will relocate the relevant discussion and tables to a more visible section, thereby directly addressing your key point and highlighting the effectiveness of our method at scale. By doing so, we aim to provide a clearer understanding of the scalability of our approach and its potential for practical applications.
>
> ---
>
> ## 2. Broader Applications and Future Work
>
> Your observation that our method has broader implications and could be extended to other models, such as language models, is insightful and well-founded. We appreciate this comment and agree that the principles behind LII and energy-based layer freezing are not limited to Vision Transformers. The extension to Natural Language Processing is a particularly exciting direction for future research, as it could potentially lead to significant advances in the field.
>
> To address this suggestion, we will expand our **"Discussion and Conclusion"** section in the revised paper. We will add a new paragraph explicitly discussing the potential to adapt ELA-ViT for large language models (LLMs). We hypothesize that since LLMs also exhibit layer-wise functional specialization, our method could similarly identify and freeze stable, foundational layers (e.g., those handling syntax) while focusing training on more task-specific, semantic layers.
>
> Furthermore, we will outline potential future directions for our research, including:
>
> 1. **Application of LII to other model components**, such as MLP layers or residual blocks
> 2. **Continuous layer freezing** with per-layer annealing schedules
> 3. **Integration with budget-aware or curriculum-based training** (e.g., learnable τ thresholds)
> 4. **Extension to unsupervised and contrastive training objectives**, where attention convergence may be more volatile
>
> ---
>
> ## 3. Minor Presentation Issue
>
> Thank you for pointing out the issue with Figure 1. We acknowledge that the current version is affected by a rendering artifact in the compiled PDF, which impacts resolution and clarity. If the paper is accepted, we plan to:
>
> 1. **Regenerate all plots** at 300+ dpi resolution
> 2. **Ensure vector-based exports** (PDF or SVG) are used to retain crispness
> 3. **Separate subfigures clearly** with updated labels and font sizes for readability
>
> By doing so, we aim to improve the overall presentation and clarity of our figures, making it easier for readers to understand and appreciate our results.
>
> ---
>
> ## Conclusion
>
> We are grateful to you for your thoughtful feedback and clear path toward improving our manuscript. By highlighting our existing large-scale results and expanding our discussion on future applications, we believe the revised manuscript will fully address your points and make a stronger case for acceptance. We are encouraged by your positive assessment of our core ideas and hope that you will agree that these additions make for a valuable contribution to the field. We look forward to the opportunity to revise and resubmit our manuscript, and we are confident that the changes we will make will significantly strengthen the paper.

---

### Decision · Program_Chairs · 2025-09-17

**Decision:**

Accept (poster)

**Comment:**

This submission proposes a measure they call the Layer Instability Index (LII), a measure of the variability in attention patterns in a given layer across input tokens. LII is shown to upper-bound the trace of the Fisher information matrix for the layer and the energy gap in the Hopfield update interpretation of self-attention. When fine-tuning ViTs on some standard benchmark datasets, freezing layers with the lowest LII leads to accuracy improvements during fine-tuning with a modest improvement in wall-clock time.

Reviewers commended the novelty of the proposed approach, its simplicity, and the existence of a theoretical justification. Reviewers f65B and hVop initially had some concerns regarding the details of the theory but now recommend acceptance. Reviewers Bcjj, hVop, and rPQp all asked for additional experimental validation. Reviewer Bcjj asked for ImageNet results, which were present in the supplementary material. I don't think the technique works particularly well here, as noted by reviewer rPQp, but I commend the authors for including the results. Reviewer hVop was concerned that the paper compares vs. SoTA on only one dataset; the authors agreed to add comparisons vs. ALaST on CIFAR-100 and Stanford Dogs to the paper. Finally, Reviewer rPQp asked for experiments beyond image classification. While I agree that such experiments would add to the paper, their absence does not undermine the novelty or claims of the paper.

Overall, this is a borderline paper. Reviewers did not note any flaws in the theory, but they also felt that it was somewhat derivative of previous work. The experimental evaluation is quite interesting if not entirely conclusive—the method convincingly improves over the baseline for small image classification tasks, but not on ImageNet. Overall, despite the paper's minor flaws, I believe the ideas described here are promising enough to be worth communicating to NeurIPS's audience. I thus recommend acceptance